# DO-GAN: A DOUBLE ORACLE FRAMEWORK FOR GENERATIVE ADVERSARIAL NETWORKS

## ABSTRACT

In this paper, we propose a new approach to train Generative Adversarial Networks (GANs) where we deploy a double-oracle framework using the generator and discriminator oracles. GAN is essentially a two-player zero-sum game between the generator and the discriminator. Training GANs is challenging as a pure Nash equilibrium may not exist and even finding the mixed Nash equilibrium is difficult as GANs have a large-scale strategy space. In DO-GAN, we extend the double oracle framework to GANs. We first generalize the player strategies as the trained models of generator and discriminator from the best response oracles. We then compute the meta-strategies using a linear program. Next, we prune the weakly-dominated player strategies to keep the oracles from becoming intractable. We apply our framework to established architectures such as vanilla GAN, Deep Convolutional GAN, Spectral Normalization GAN and Stacked GAN. Finally, we conduct evaluations on MNIST, CIFAR-10 and CelebA datasets and show that DO-GAN variants have significant improvements in both subjective qualitative evaluation and quantitative metrics, compared with their respective GAN architectures.

## 1 INTRODUCTION

Generative Adversarial Networks (GANs) (Goodfellow et al., 2014) have been applied in various domains such as image and video generation, image-to-image translation and text-to-image synthesis (Liu et al., 2017; Reed et al., 2016). Various architectures are proposed to generate more realistic samples (Radford et al., 2015; Mirza & Osindero, 2014; Pu et al., 2016) as well as regularization techniques (Arjovsky et al., 2017; Miyato et al., 2018b). From the game-theoretic perspective, GANs can be viewed as a two-player game where the generator samples the data and the discriminator classifies the data as real or generated. The two networks are alternately trained to maximize their respective utilities until convergence corresponding to a pure Nash Equilibrium (NE).

However, pure NE cannot be reliably reached by existing algorithms as pure NE may not exist (Farnia & Ozdaglar, 2020; Mescheder et al., 2017). This also leads to unstable training in GANs depending on the data and the hyperparameters. Therefore, mixed NE is a more suitable solution concept (Hsieh et al., 2019). Several recent works propose mixture architectures with multiple generators and discriminators that consider mixed NE such as MIX+GAN (Arora et al., 2017) and MGAN (Hoang et al., 2018). MIX+GAN and MGAN cannot guarantee to converge to mixed NE. Mirror-GAN (Hsieh et al., 2019) finds the mixed NE by sampling over the infinite-dimensional strategy space and proposes provably convergent proximal methods. However, the sampling approach may not be efficient as mixed NE may only have a few strategies in the support set.

Double Oracle (DO) algorithm (McMahan et al., 2003) is a powerful framework to compute mixed NE in large-scale games. The algorithm starts with a restricted game with a small set of actions and solves it to get the NE strategies of the restricted game. The algorithm then computes players' best-responses using oracles to the NE strategies and add them into the restricted game for the next iteration. DO framework has been applied in various disciplines (Jain et al., 2011; Bošanský et al., 2013), as well as Multi-agent Reinforcement Learning (MARL) settings (Lanctot et al., 2017).

Inspired by the successful applications of DO framework, we, for the first time, propose a Double Oracle Framework for Generative Adversarial Networks (DO-GAN). This paper presents four key contributions. First, we treat the generator and the discriminator as players and obtain the best responses from their oracles and add the utilities to a meta-matrix. Second, we propose a linear

program to obtain the probability distributions of the players' pure strategies (meta-strategies) for the respective oracles. The linear program computes an exact mixed NE of the meta-matrix game in polynomial time. Third, we propose a pruning method for the support set of best response strategies to prevent the oracles from becoming intractable as there is a risk of the meta-matrix growing very large with each iteration of oracle training. Finally, we provide comprehensive evaluation on the performance of DO-GAN with different GAN architectures using both synthetic and real-world datasets. Experiment results show that DO-GAN variants have significant improvements in terms of both subjective qualitative evaluation and quantitative metrics.

## 2   RELATED WORKS

In this section, we briefly introduce existing GAN architectures, double oracle algorithm and its applications such as policy-state response oracles that are related to our work.

**GAN Architectures.**   Various GAN architectures have been proposed to improve the performance of GANs. Deep Convolutional GAN (DCGAN) (Radford et al., 2015) replaces fully-connected layers in the generator and the discriminator with deconvolution layer of Convolutional Neural Networks (CNN). Weight normalization techniques such as Spectral Normalization GAN (SNGAN) (Miyato et al., 2018a) stabilize the training of the discriminator and reduce the intensive hyperparameters tuning. There are also multi-model architectures such as Stacked Generative Adversarial Networks (SGAN) (Huang et al., 2017) that consist of a top-down stack of generators and a bottom-up discriminator network. Each generator is trained to generate lower-level representations conditioned on higher-level representations that can fool the corresponding representation discriminator. Training GANs is very hard and unstable as pure NE for GANs might not exist and cannot be reliably reached by the existing approaches (Mescheder et al., 2017). Considering mixed NE, MIX+GAN (Arora et al., 2017) maintains a mixture of generators and discriminators with the same network architecture but have their own trainable parameters. However, training a mixture of networks without parameter sharing makes the algorithm computationally expensive. Mixture Generative Adversarial Nets (MGAN) (Hoang et al., 2018) propose to capture diverse data modes by formulating GAN as a game between a classifier, a discriminator and multiple generators with parameter sharing. However, MIX+GAN and MGAN cannot converge to mixed NE. Mirror-GAN (Hsieh et al., 2019) finds the mixed NE by sampling over the infinite-dimensional strategy space and proposes provably convergent proximal methods. The sampling approach may be inefficient to compute mixed NE as the mixed NE may only have a few strategies with positive probabilities in the infinite strategy space.

**Double Oracle Algorithm.**   Double Oracle (DO) algorithm starts with a small restricted game between two players and solves it to get the player strategies at NE of the restricted game. The algorithm then exploits the respective best response oracles for additional strategies of the players. The DO algorithm terminates when the best response utilities are not higher than the equilibrium utility of the current restricted game, hence, finding the NE of the game without enumerating the entire strategy space. Moreover, in two-player zero-sum games, DO converges to a min-max equilibrium (McMahan et al., 2003). DO framework is used to solve large-scale normal-form and extensive-form games such as security games (Tsai et al., 2012; Jain et al., 2011), poker games (Waugh et al., 2009) and search games (Bosansky et al., 2012). DO framework is also used in MARL settings (Lanctot et al., 2017; Muller et al., 2020). Policy-Space Response Oracles (PSRO) generalize the double oracle algorithm in a multi-agent reinforcement learning setting (Lanctot et al., 2017). PSRO treats the players' policies as the best responses from the agents' oracles, builds the meta-matrix game and computes the mixed NE but it uses Projected Replicator Dynamics that update the changes in the probability of each player's policy at each iteration. Since the dynamics need to simulate the update for several iterations, the use of dynamics takes a longer time to compute the meta-strategies and does not guarantee to compute an exact NE of the meta-matrix game. However, in DO-GAN, we can use a linear program to compute the players' meta-strategies in polynomial time since GAN is a two-player zero-sum game (Schrijver, 1998).

## 3 PRELIMINARIES

In this section, we mathematically explain the preliminary works that are needed to explain our DO-GAN approach including generative adversarial networks and game theory concepts such as normal-form game and double oracle algorithm.

### 3.1 GENERATIVE ADVERSARIAL NETWORKS

Generative Adversarial Networks (GANs) (Goodfellow et al., 2014) have become one of the dominant methods for fitting generative models to complicated real-life data. GANs are deep neural net architectures comprised of two neural networks trained in an adversarial manner to generate data that resembles a distribution. The first neural network, a generator $G$, is given some random distribution $p_{\mathbf{z}}(\mathbf{z})$ on the input noise $\mathbf{z}$ and a real data distribution $p_{data}(\mathbf{x})$ on training data $\mathbf{x}$. The generator is supposed to generate as close as possible to $p_{data}(\mathbf{x})$. The second neural network, a discriminator $D$, is to discriminate between two different classes of data (real or fake) from the generator.

Let the generator's differentiable function be denoted as $G(\mathbf{z}, \pi_g)$ and similarly $D(\mathbf{x}, \pi_d)$ for the discriminator, where $G$ and $D$ are two neural networks with parameters $\pi_g$ and $\pi_d$. Thus, $D(\mathbf{x})$ represents the probability that $\mathbf{x}$ comes from the real data. The generator loss $L_G$ and the discriminator loss $L_D$ are defined as:

$$L_D = \mathbb{E}_{\mathbf{x} \sim p_{data}(\mathbf{x})}[-\log D(\mathbf{x})] + \mathbb{E}_{\mathbf{z} \sim p_{\mathbf{z}}(\mathbf{z})}[-\log(1 - D(G(\mathbf{z})))], \tag{1}$$

$$L_G = \mathbb{E}_{\mathbf{z} \sim p_{\mathbf{z}}(\mathbf{z})}[\log(1 - D(G(\mathbf{z})))]. \tag{2}$$

GAN is then set up as a two-player zero-sum game between $G$ and $D$ as follows:

$$\min_G \max_D \mathbb{E}_{\mathbf{x} \sim p_{data}(\mathbf{x})}[\log D(\mathbf{x})] + \mathbb{E}_{\mathbf{z} \sim p_{\mathbf{z}}(\mathbf{z})}[\log(1 - D(G(\mathbf{z})))]. \tag{3}$$

During training, the parameters of $G$ and $D$ are updated alternately until we reach the global optimal solution $D(G(\mathbf{z})) = 0.5$. Next, we let $\Pi_g$ and $\Pi_d$ be the set of parameters for $G$ and $D$, considering the set of probability distributions $\sigma_g$ and $\sigma_d$, the mixed strategy formulation (Hsieh et al., 2019) is:

$$\min_{\sigma_g} \max_{\sigma_d} \mathbb{E}_{\pi_d \sim \sigma_d} \mathbb{E}_{\mathbf{x} \sim p_{data}(\mathbf{x})}[\log D(\mathbf{x}, \pi_d)] + \mathbb{E}_{\pi_d \sim \sigma_d} \mathbb{E}_{\pi_g \sim \sigma_g} \mathbb{E}_{\mathbf{z} \sim p_{\mathbf{z}}(\mathbf{z})}[\log(1 - D(G(\mathbf{z}, \pi_g), \pi_d))]. \tag{4}$$

Similarly to GANs, DCGAN, SNGAN and SGAN can also be viewed as two-player zero-sum games with mixed strategies of the players. DCGAN modifies the vanilla GAN by replacing fully-connected layers with the convolutional layers. SGAN trains multiple generators and discriminators using the loss as a linear combination of 3 loss terms: adversarial loss, conditional loss and entropy loss.

### 3.2 NORMAL FORM GAME AND DOUBLE ORACLE ALGORITHM

A normal-form game is a tuple $(\Pi, U, n)$ where $n$ is the number of players, $\Pi = (\Pi_1, \ldots, \Pi_n)$ is the set of strategies for each player $i \in N$, where $N = \{1, \ldots, n\}$ and $U : \Pi \rightarrow R^n$ is a payoff table of utilities $R$ for each joint policy played by all players. Each player chooses the strategy to maximize own expected utility from $\Pi_i$, or by sampling from a distribution over the set of strategies $\sigma_i \in \Delta(\Pi_i)$. We can use linear programming, fictitious play (Berger, 2007) or regret minimization (Roughgarden, 2010) to compute the probability distribution over players' strategies.

In the Double Oracle (DO) algorithm (McMahan et al., 2003), there are two best response oracles for the row and column player respectively. The algorithm creates restricted games from a subset of strategies at the point of each iteration $t$ for row and column players, i.e., $\Pi_r^t \subset \Pi_r$ and $\Pi_c^t \subset \Pi_c$ as well as a meta-matrix $U^t$ at the $t^{th}$ iteration. We then solve the meta-matrix to get the probability distributions on $\Pi_r^t$ and $\Pi_c^t$. Given a probability distribution $\sigma_c$ of the column player strategies, $BR_r(\sigma_c)$ gives the row player's best response to $\sigma_c$. Similarly, given probability distribution $\sigma_r$ of the row player's strategies, $BR_c(\sigma_r)$ is the column player's best response to $\sigma_r$. The best responses are added to the restricted game for the next iteration. The algorithm terminates when the best response utilities are not higher than the equilibrium utility of current restricted game. Although in the worst-case, the entire strategy space may be added to the restricted game, DO is guaranteed to converge to mixed NE in two-player zero-sum games. DO is also extended to the multi-agent reinforcement learning in PSRO (Lanctot et al., 2017) to approximate the best responses to the mixtures of agents' policies, and compute the meta-strategies for the policy selection.

Table 1: Comparison of Terminologies between Game Theory and GAN

| Game Theory terminology | GAN terminology |
|---|---|
| player | generator/ discriminator |
| strategy | the parameter setting of generator/ discriminator
e.g. $\pi_g$ and $\pi_d$ |
| policy | the sequence of parameters (strategies) till epoch $t$
e.g. $(\pi_g^1, \pi_g^2, ..., \pi_g^t)$
Note: Not used in DO-GAN. |
| game | the minmax game between generator and discriminator |
| meta-game/ meta-matrix | the minmax game between generator & discriminator
with their respective set of strategies at epoch $t$ of DO framework |
| meta-strategy | the mixed NE strategy of generator/discriminator at epoch $t$ |

## 4 DO-GAN: DOUBLE ORACLE FRAMEWORK FOR GAN

As discussed in previous sections, computing mixed NE for GANs is challenging as there is an extremely large number of pure strategies, i.e., possible parameter settings of the generator and discriminator networks. Thus, we propose a double oracle framework for GANs (DO-GAN) to compute the mixed NE efficiently. DO-GAN builds a restricted meta-matrix game between the two players and computes the mixed NE of the meta-matrix game, then DO-GAN iteratively adds more generators and discriminators into the meta-matrix game until termination.

### 4.1 GENERAL FRAMEWORK OF DO-GAN

GAN can be translated as a two-player zero-sum game between the generator player $g$ and the discriminator player $d$. To compute the mixed NE of GANs, at iteration $t$, DO-GAN creates a restricted meta-matrix game $U^t$ with the trained generators and discriminators as the strategies of the two players, where the generators and discriminators are parameterized by $\pi_g \in \mathcal{G}$ and $\pi_d \in \mathcal{D}$. We use $U^t(\pi_g, \pi_d)$ to denote the generator player's payoff when playing $\pi_g$ against $\pi_d$, which is defined as $L_D$. Since GAN is zero-sum, the discriminator player's payoff is $-U^t(\pi_g, \pi_d)$. We define $\sigma_g^t$ and $\sigma_d^t$ as the mixed strategies of generator player and discriminator player, respectively. With a slight abuse of notation, we define the generator player's expected utility of the mixed strategies $\langle \sigma_g^t, \sigma_d^t \rangle$ as $U^t(\sigma_g^t, \sigma_d^t) = \sum_{\pi_g \in \mathcal{G}} \sum_{\pi_d \in \mathcal{D}} \sigma_g^t(\pi_g) \cdot \sigma_d^t(\pi_d) \cdot U^t(\pi_g, \pi_d)$. We use $\langle \sigma_g^{t*}, \sigma_d^{t*} \rangle$ to denote the mixed NE of the restricted meta-matrix game $U^t$. We solve $U^t$ to obtain the mixed NE, compute the best responses and add them into $U^t$ for the next iteration. Figure 1 presents an illustration of DO-GAN and Algorithm 1 describes the overview of the framework.

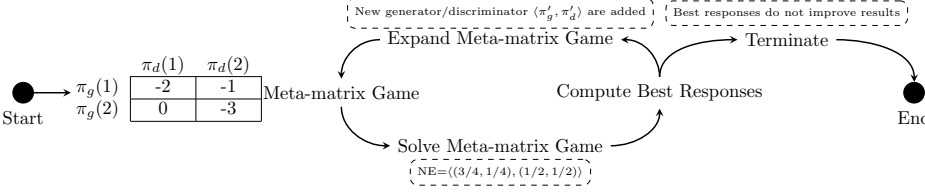

Figure 1: An illustration of DO-GAN. Figure adapted from (Lanctot et al., 2017).

Our algorithm starts by initializing two empty arrays $\mathcal{G}$ and $\mathcal{D}$ to store multiple generators and discriminators (line 1). We train the first $\pi_g$ and $\pi_d$ with the canonical training procedure of GANs (line 2). We store the parameters of the trained models in the two arrays $\mathcal{G}$ and $\mathcal{D}$ (line 3), compute the adversarial loss $L_D$ and add it to the meta-matrix $U^0$ (line 4). We initialize the meta-strategies

$\sigma_g^{0*} = [1]$ and $\sigma_d^{0*} = [1]$ since there is only one pair of generator and discriminator available (line 5). For each epoch, we use `generatorOracle()` and `discriminatorOracle()` to obtain the best responses $\pi'_g$ and $\pi'_d$ to $\sigma_d^{t*}$ and $\sigma_g^{t*}$ via Adam Optimizer, respectively, then add them into $\mathcal{G}$ and $\mathcal{D}$ (lines 7-10). We then augment $U^{t-1}$ by adding $\pi'_g$ and $\pi'_d$ and calculating $U^t(\pi'_g, \pi'_d)$ to obtain $U^t$ and compute the missing entries (line 11). We compute the missing payoff entries $U^t(\pi'_g, \pi_d), \forall \pi_d \in \mathcal{D}$ and $U^t(\pi_g, \pi'_d), \forall \pi_g \in \mathcal{G}$ by sampling a few batches of training data. After that, we compute the mixed NE $\langle \sigma_g^{t*}, \sigma_d^{t*} \rangle$ of $U^t$ with linear programming (line 12). The algorithm terminates if the criteria described in Algorithm 2 is satisfied (line 13). We also prune the support strategy set of the players as described in Algorithm 3 (line 14) to avoid $\mathcal{G}$ and $\mathcal{D}$ becoming intractable.

In `generatorOracle()`, we train $\pi'_g$ to obtain the best response against $\sigma_d^{t*}$, i.e., $U^t(\pi'_g, \sigma_d^{t*}) \geq U^t(\pi_g, \sigma_d^{t*}), \forall \pi_g \in \Pi_g$. Similarly, in `discriminatorOracle()`, we train $\pi'_d$ to obtain the best response against $\sigma_g^{t*}$, i.e., $U^t(\sigma_g^{t*}, \pi'_d) \geq U^t(\sigma_g^{t*}, \pi_d), \forall \pi_d \in \Pi_d$. Full details of generator oracle and discriminator oracle can be found in Appendix A.

---

**Algorithm 1:** Double Oracle Framework for GAN (`DO-GAN`)

1  Initialize generator and discriminator arrays $\mathcal{G} = \emptyset$ and $\mathcal{D} = \emptyset$;
2  Train generator & discriminator to get the first $\pi_g$ and $\pi_d$;
3  $\mathcal{G} \leftarrow \mathcal{G} \cup \{\pi_g\}; \mathcal{D} \leftarrow \mathcal{D} \cup \{\pi_d\}$;
4  Compute the adversarial loss $L_D$ and add it to meta-matrix $U^0$;
5  Initialize $\sigma_g^{0*} = [1]$ and $\sigma_d^{0*} = [1]$;
6  **for** *epoch* $t \in \{1, 2, ...\}$ **do**
7     $\pi'_g \leftarrow$ `generatorOracle`$(\sigma_d^{t*}, \mathcal{D})$;
8     $\mathcal{G} \leftarrow \mathcal{G} \cup \{\pi'_g\}$;
9     $\pi'_d \leftarrow$ `discriminatorOracle`$(\sigma_g^{t*}, \mathcal{G})$;
10    $\mathcal{D} \leftarrow \mathcal{D} \cup \{\pi'_d\}$;
11    Augment $U^{t-1}$ with $\pi'_g$ and $\pi'_d$ to obtain $U^t$ and compute missing entries;
12    Compute mixed NE $\langle \sigma_g^{t*}, \sigma_d^{t*} \rangle$ for $U^t$ with linear program;        // Section 4.2
13    **if** `TerminationCheck`$(U^t, \sigma_g^{t*}, \sigma_d^{t*})$ **then break**;        // Section 4.3
14    `PruneMetaMatrix`$(U^t, \sigma_g^{t*}, \sigma_d^{t*})$;        // Section 4.4

---

**Algorithm 2:** `TerminationCheck`$(U^t, \sigma_g^{t*}, \sigma_d^{t*})$

// $U^t$ is of size $m \times n$
// $|G| = m, |D| = n$
1  Compute $U^t(\sigma_g^{t*}, \sigma_d^{t*})$;
2  Compute $U^t(\sigma_g^{t*}, \mathcal{D}[n])$;
3  Compute $U^t(\mathcal{G}[m], \sigma_d^{t*})$;
4  $genInc = U^t(\mathcal{G}[m], \sigma_d^{t*}) - U^t(\sigma_g^{t*}, \sigma_d^{t*})$;
5  $disInc = -U^t(\sigma_g^{t*}, \mathcal{D}[n]) - (-U^t(\sigma_g^{t*}, \sigma_d^{t*}))$;
6  **if** $genInc < \epsilon$ && $-disInc < \epsilon$ **then**
7     **return** True
8  **else return** False ;

---

**Algorithm 3:** `PruneMetaMatrix`$(U^t, \sigma_g^{t*}, \sigma_d^{t*})$

1  // I stores indices to be pruned from $\mathcal{G}$ and $\mathcal{D}$
   // G stores models to be pruned from $\mathcal{G}$ and $\mathcal{D}$
2  $I_g = \emptyset; I_d = \emptyset$
3  $K_g = \emptyset; K_d = \emptyset$ **if** $|\mathcal{G}| > s$ **then**
4     **for** $i \in \{0, \ldots, |\mathcal{G}| - 1\}$ **do**
5        **if** $\sigma_g^{t*}(\mathcal{G}[i]) == \min \sigma_g^{t*}$ **then**
         $I_g \leftarrow I_g \cup \{i\}; K_g \leftarrow K_g \cup \{\mathcal{G}[i]\}$ ;

6  **if** $|\mathcal{D}| > s$ **then**
7     **for** $j \in \{0, \ldots, |\mathcal{D}| - 1\}$ **do**
8        **if** $\sigma_d^{t*}(\mathcal{D}[j]) == \min \sigma_d^{t*}$ **then**
         $I_d \leftarrow I_d \cup \{j\}; K_d \leftarrow K_d \cup \{\mathcal{D}[j]\}$ ;

9  $\mathcal{G} \leftarrow \mathcal{G} \setminus K_g; \mathcal{D} \leftarrow \mathcal{D} \setminus K_d$;
10 $U \leftarrow J_{I_g, m} \cdot U^t \cdot J_{I_d, n}^T$

---

### 4.2 LINEAR PROGRAM OF META-MATRIX GAME

Since the current restricted meta-matrix game $U^t$ is a zero-sum game, we can use a linear program to compute the mixed NE in polynomial time (Schrijver, 1998). Given the generator player $g$'s mixed strategy $\sigma_g^t$, the discriminator player $d$ will play strategies that minimize the expected utility of $g$. Thus, the mixed NE strategy for the generator player $\sigma_g^{t*}$ is to maximize the worst-case expected utility, which is obtained by solving the following linear program:

$$\sigma_g^{t*} = \arg\max_{\sigma_g^t} \{v : \sigma_g^t \geq 0, \sum_{i \in \mathcal{G}} \sigma_g^t(i) = 1, U^t(\sigma_g^t, \pi_d) \geq v, \forall \pi_d \in \mathcal{D}\}. \quad (5)$$

Similarly, we can obtain the mixed NE strategy for the discriminator $\sigma_d^{t*}$ by solving a linear program that maximizes the worst-case expected utility of the discriminator player. Therefore, we obtain the mixed NE $\langle \sigma_g^{t*}, \sigma_d^{t*} \rangle$ of the restricted meta-matrix game $U^t$.

### 4.3 Termination Check

DO terminates the training by checking whether the best response $\pi_g'$ (or $\pi_d'$) is in the support set $\mathcal{G}$ (or $\mathcal{D}$) (Jain et al., 2011), but we cannot apply this approach to DO-GAN as GAN has infinite-dimensional strategy space (Hsieh et al., 2019). Hence, we terminate the training if the best responses cannot bring a higher utility to the two players than the entries of the current support sets, as discussed in (Lanctot et al., 2017; Muller et al., 2020). Specifically, we first compute $U^t(\sigma_g^{t*}, \sigma_d^{t*})$ and the expected utilities for new generator and discriminator $U^t(\mathcal{G}[m], \sigma_d^{t*}), U^t(\sigma_g^{t*}, \mathcal{D}[n])$ (line 1-3). Then, we calculate the utility increment (lines 4-5) and returns **True** if both $U^t(\mathcal{G}[m], \sigma_d^{t*})$ and $U^t(\sigma_g^{t*}, \mathcal{D}[n])$ cannot bring a higher utility than $U^t(\sigma_g^{t*}, \sigma_d^{t*})$ by $\epsilon$ (lines 6-8).

### 4.4 Pruning Meta-matrix

As the meta-matrix grows with every epoch of DO, there is a risk that the support strategy set becomes very large and $\mathcal{G}$ and $\mathcal{D}$ become intractable. To avoid this, we adapt the greedy pruning algorithm from (Cheng & Wellman, 2007), as depicted in Algorithm 3. When either $|\mathcal{G}|$ or $|\mathcal{D}|$ is greater than the limit of the support set size $s$, we prune at least one strategy with the least probability, which is the strategy that contributes the least to the player's winning. Specifically, we define $J_{I,b}$ where $I$ is the set of row numbers to be removed, $b$ is the total rows of a matrix. To remove the $2^{nd}$ row of a matrix having 3 rows, we define $I = \{1\}, b = 3$ and $J_{\{1\},3} = \begin{pmatrix} 1 & 0 & 0 \\ 0 & 0 & 1 \end{pmatrix}$. If $|\mathcal{G}| > s$, at least one strategy with minimum probability is pruned from $\mathcal{G}$, similarly for $\mathcal{D}$ (lines 3-9). Finally, we prune the meta-matrix using matrix multiplication (line 10).

## 5 Experiments

We conduct our experiments on a machine with Xeon(R) CPU E5-2683 v3@2.00GHz and $4\times$ Tesla v100-PCIE-16GB running Ubuntu operating system. We evaluate the double oracle framework for some established GAN architectures such as vanilla GAN (Goodfellow et al., 2014), DCGAN (Radford et al., 2015), SNGAN (Miyato et al., 2018a) and SGAN (Huang et al., 2017). We adopt the parameter settings and criterion of the GAN architectures as published. We set $s = 10$ unless mentioned otherwise. We compute the mixed NE of the meta-matrix game with Nashpy[1]. The evaluation details are shown in Appendix B.

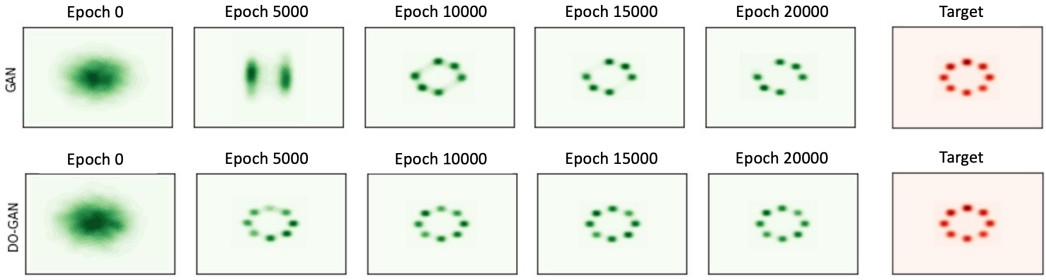

Figure 2: Comparison of GAN and DO-GAN on 2D synthetic dataset

### 5.1 Evaluation on Synthetic 2D Gaussian Mixture Dataset

To illustrate the effectiveness of the architecture, we train a double oracle framework with the simple vanilla GAN architecture on a 2D mixture of 8 Gaussian mixture components with cluster standard

---

[1]`https://nashpy.readthedocs.io/en/stable/index.html`

deviation 0.1 which follows the experiment by (Metz et al., 2017). Figure 2 shows the evolution of 512 samples generated by GAN and DO-GAN through 20000 epochs. The goal of GAN and DO-GAN is to correctly generate samples at 8 modes as shown in the target. The results show that GAN can only identify 6 out of 8 modes of the synthetic Gaussian data distribution, while the DO-GAN can obtain all the 8 modes of the distribution. Furthermore, DO-GAN takes shorter time (less than 5000 epochs) to identify all 8 modes of the data distribution. We present a more detailed evolution of data samples through the training process on 2D Gaussian Mixtures in Appendix C.

**Ablations.** We also varied the support set size of the training $s = 5, 10, 15$ and recorded the computation time as discussed in Appendix D. We found that the training cannot converge when $s = 5$ and takes a long time when $s = 15$. Thus, we chose $s = 10$ for the training process.

## 5.2 EVALUATION ON REAL-WORLD DATASETS

We evaluate the performance of the double oracle framework which takes several established GAN architectures as the backbone as discussed in Appendix G, i.e., GAN (Goodfellow et al., 2014), DCGAN (Radford et al., 2015) and SGAN (Huang et al., 2017) with convolutional layers for the deep neural networks of GAN as well as SNGAN (Miyato et al., 2018a) which uses normalization techniques. We run experiments on MNIST (LeCun & Cortes, 2010), CIFAR-10 (Krizhevsky et al., 2009) and CelebA (Liu et al., 2015) datasets. MNIST contains 60,000 samples of handwritten digits with images of $28 \times 28$. CIFAR-10 contains $50,000$ training images of $32 \times 32$ of 10 classes. CelebA is a large-scaled face dataset with more than 200K images of size $128 \times 128$.

### 5.2.1 QUALITATIVE EVALUATION

We choose the CelebA dataset for the qualitative evaluation since the training images contain noticeable artifacts (aliasing, compression, blur) that make the generator difficult to produce perfect and faithful images. We compare the performance of DO-DCGAN, DO-SNGAN and DO-SGAN with their counterparts [2]. SNGAN which is trained for 40 epochs with termination $\epsilon$ of $5 \times 10^{-5}$ for DO-SNGAN where other architectureas are trained for 25 epochs with termination $\epsilon$ of $5 \times 10^{-5}$ for the double oracle variants. The generated CelebA images of DCGAN and DO-DCGAN are shown in Figure 3, where we find that DCGAN suffers mode-collapse, while DO-DCGAN does not. We also present the generated images of SNGAN vs DO-SNGAN and SGAN vs DO-SGAN using fixed noise at different training epochs in Figure 4 and 5. From the results, we can see that SNGAN, SGAN, DO-SNGAN and DO-SGAN are able to generate various faces, i.e., no mode-collapse. Judging from subjective visual quality, we find that DO-SNGAN and DO-SGAN are able to generate plausible images faster than SNGAN and SGAN during training, i.e., 17 epochs for DO-SGAN and 20 epochs for SGAN. More experiment results on CIFAR-10 can be found in Appendix E.

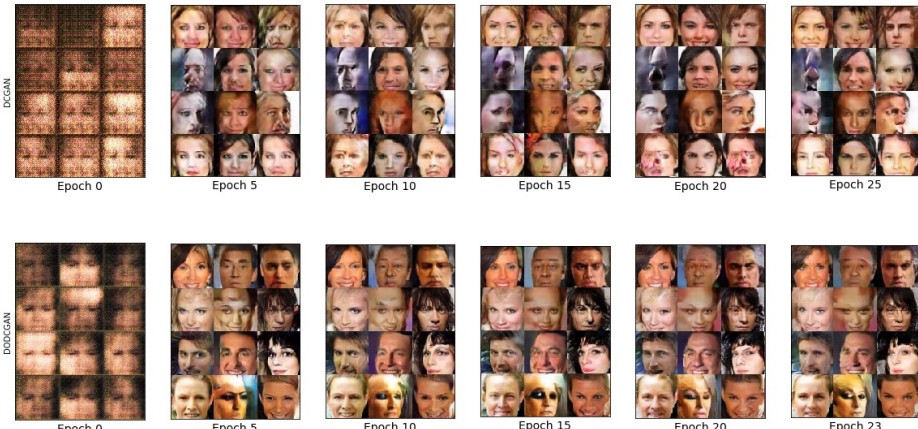

Figure 3: Training images with fixed noise for DCGAN and DO-DCGAN until termination.

---

[2]We do not evaluate the performance of vanilla GAN and its DO variant on CelebA dataset since DCGAN and SGAN outperform vanilla GAN in image generation tasks (Radford et al., 2015)

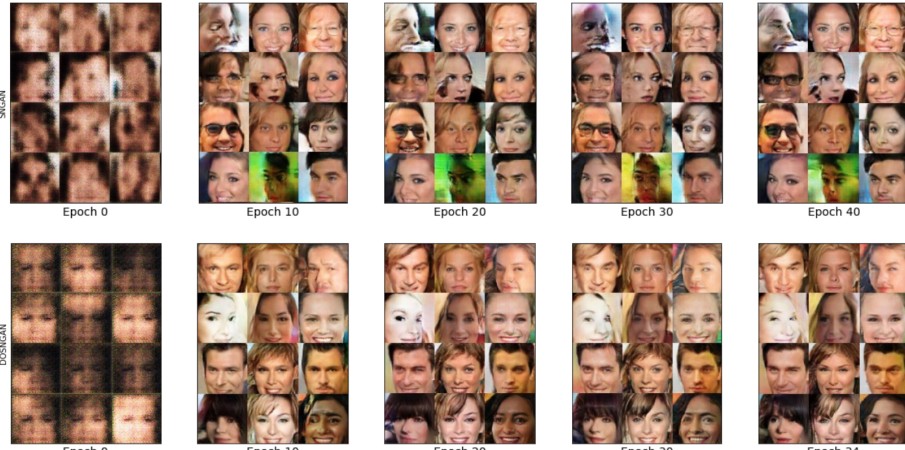

Figure 4: Training images with fixed noise for SNGAN and DO-SNGAN until termination.

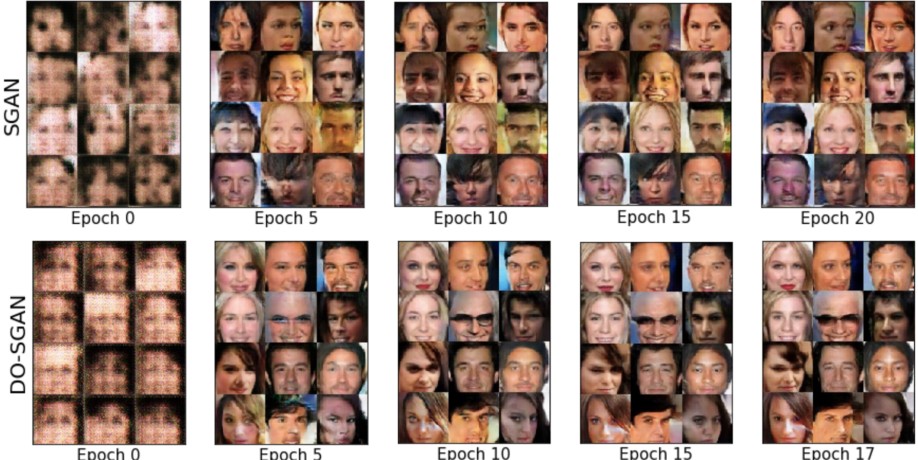

Figure 5: Training images with fixed noise for SGAN and DO-SGAN until termination.

### 5.2.2 QUANTITATIVE EVALUATION

In this section, we evaluate the performance of various architectures by quantitative metrics.

**Inception Score.** We first leverage the Inception Score (IS) (Salimans et al., 2016) by using Inception_v3 (Szegedy et al., 2016) as the inception model. To compute the inception score, we first compute the Kullback-Leibler (KL) divergence for all generated images and use the equation $IS = \exp(\mathbb{E}_{\mathbf{x}}[KL(D(p(y|\mathbf{x}) \parallel p(y)))])$ where $p(y)$ is the conditional label distributions for the images in the split and $p(y|\mathbf{x})$ is that of the image $\mathbf{x}$ estimated by the reference inception model. Inception score evaluates the quality and diversity of all generated images rather than the similarity to the real data from the test set.

**FID Score.** Fréchet Inception Distance (FID) measures the distance between the feature vectors of real and generated images using Inception_v3 model (Heusel et al., 2017). Here, we let $p$ and $q$ be the distributions of the representations obtained by projecting real and generated samples to the last hidden layer of Inception model. Assuming that $p$ and $q$ are the multivariate Gaussian distributions, FID measures the 2-Wasserstein distance between the two distributions. Hence, FID Score can capture the similarity of generated images to real ones better than the Inception score.

Table 2: Inception scores (higher is better) and FID scores (lower is better). The mean and standard deviation are drawn from running 10 splits on 10000 generated images. The magenta values are the improvements of the DO-GAN variants compared with their counterparts.

| | Inception Score | | FID Score | |
|---|---|---|---|---|
| | MNIST | CIFAR-10 | CIFAR-10 | CelebA |
| GAN | $1.04 \pm 0.05$ | $3.84 \pm 0.09$ | 71.44 | - |
| DCGAN | $1.26 \pm 0.05$ | $6.32 \pm 0.05$ | 37.66 | 10.92 |
| SNGAN | $1.35 \pm 0.11$ | $7.58 \pm 0.12$ | 25.5 | 7.62 |
| SGAN | $1.39 \pm 0.09$ | $8.62 \pm 0.12$ | 24.83 | 6.98 |
| MIX+DCGAN | - | $7.72 \pm 0.09$ | - | - |
| MGAN | - | $8.33 \pm 0.10$ | 26.7 | - |
| DO-GAN | $1.39 \pm 0.09 \, (+0.35)$ | $7.20 \pm 0.16 \, (+3.36)$ | $31.44 \, (-40.00)$ | - |
| DO-DCGAN | $1.42 \pm 0.11 \, (+0.16)$ | $7.86 \pm 0.14 \, (+1.54)$ | $22.25 \, (-15.41)$ | $7.11 \, (-3.81)$ |
| DO-SNGAN | $\mathbf{1.42 \pm 0.07} \, (+0.07)$ | $8.55 \pm 0.08 \, (+0.97)$ | $18.20 \, (-7.30)$ | $6.92 \, (-0.70)$ |
| DO-SGAN | $1.42 \pm 0.09 \, (+0.03)$ | $\mathbf{8.69 \pm 0.10} \, (+0.07)$ | $\mathbf{16.56} \, (-8.27)$ | $\mathbf{6.32} \, (-0.66)$ |

Note: MIX+DCGAN and MGAN results are directly copied from (Arora et al., 2017; Hoang et al., 2018).

**Results.** The results are shown in Table 2. In CIFAR-10 dataset, DO-GAN, DO-DCGAN and DO-SNGAN obtain much better results ($7.2 \pm 0.16$, $7.86 \pm 0.14$ and $8.55 \pm 0.08$) than GAN, DCGAN and SNGAN ($3.84 \pm 0.09$, $6.32 \pm 0.05$ and $7.58 \pm 0.12$). However, we do not see a significant improvement in DO-SGAN compared to SGAN $8.62 \pm 0.12$ and $8.69 \pm 0.10$ since SGAN already can generate diverse images. We did not include IS for CelebA dataset as IS cannot reflect the real image quality for the CelebA, as observed in (Heusel et al., 2017). In CIFAR-10 dataset, DO-GAN, DO-DCGAN, DO-SNGAN and DO-SGAN obtain much lower FID scores (31.44, 22.25, 16.56, 18.20) respectively. The trend follows in CelebA obtaining 7.11 for DO-DCGAN while 10.92 for DCGAN, 7.62 for SNGAN while 6.92 for DO-SNGAN, 6.98 for SGAN and 6.32 for DO-SGAN respectively. Although we see a significant improvement in the quality of DO-SGAN images, FID score for DO-SGAN is affected by the distortions. According to the results, we can see that DO framework performs better than each of their original counterpart architectures. More details can be found in Appendix F.

## 6  CONCLUSION

We propose a novel double oracle framework to GANs, which starts with a restricted game and incrementally adds the best responses of the generator and the discriminator to compute the mixed NE. We then compute the players' meta-strategies by using a linear program. We also prune the support strategy set of players. We apply DO-GAN approach to established GAN architectures such as vanilla GAN, DCGAN, SNGAN and SGAN. Extensive experiments with the 2D Gaussian synthetic data set as well as real-world datasets such as MNIST, CIFAR-10 and CelebA show that DO-GAN variants have significant improvements in comparison to their respective GAN architectures both in terms of subjective image quality as well as in terms of quantitative metrics.

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

## A    FULL ALGORITHM OF DO-GAN

---

**Algorithm 4:** `GeneratorOracle`$(\sigma_d^{t*}, \mathcal{D})$

---

**1** Initialize a generator $G$ with random parameter setting $\pi_g'$;

**2 for** *iteration* $k_0 \dots k_n$ **do**

**3**    Sample noise $\mathbf{z}$;

**4**    $\pi_d =$ Sample a discriminator from $\mathcal{D}$ with $\sigma_d^{t*}$;

**5**    Initialize a discriminator $D$ with parameter setting $\pi_d$;

**6**    Update the generator $G$'s parameters $\pi_g'$ via Adam optimizer:

$$\bigtriangledown_{\pi_g'} \log\left(1 - D(G(\mathbf{z}))\right)$$

---

---

**Algorithm 5:** `DiscriminatorOracle`$(\sigma_g^{t*}, \mathcal{G})$

---

**1** Initialize a discriminator $D$ with random parameter setting $\pi_d'$;

**2 for** *iteration* $k_0 \dots k_n$ **do**

**3**    Sample a minibatch of data $\mathbf{x}$;

**4**    **for** *a minibatch* **do**

**5**       Sample noise $\mathbf{z}$;

**6**       $\pi_g =$ Sample a generator from $\mathcal{G}$ with $\sigma_g^{t*}$;

**7**       Initialize a generator $G$ with a parameter setting $\pi_g$;

**8**       Generate and add to mixture $G(\mathbf{z})$;

**9**    Update the discriminator $D$'s parameters $\pi_d'$ via Adam optimizer:

**10**

$$\bigtriangledown_{\pi_d'} \log D(\mathbf{x}) + \log\left(1 - D(G(\mathbf{z}))\right)$$

---

We train the oracles for some iterations which we denote as $k_{0,1,2,\dots}$. For experiments, we train each oracle for an epoch for the real-world datasets and 50 iterations for the 2D Synthetic Gaussian Dataset. At each iteration $t$, we sample the generators from the support set $\mathcal{G}$ with the meta-strategy $\sigma_g^{t*}$ to generate the images for evaluation. Similarly, we conduct the performance evaluation with the generators sampled from $\mathcal{G}$ with the final $\sigma_g^*$ at termination. SGAN consists of a top-down stack of GANs, e.g, for a stack of 2, Generator 1 is the first layer stacked on Generator 0 with each of them connected to Discriminator 1 and 0 respectively. Hence, in DO-SGAN, we store the meta-strategies for the Generator 0 and 1 in $\sigma_g^{t*}$ and the Discriminator 1 and 0 for $\sigma_d^{t*}$. In `GeneratorOracle()`, we first sample Discriminator 1 and 0 from discriminator distribution $\sigma_d^{t*}$ and train Generator 1 first then followed by calculating loss with Discriminator 1 and train Generator 0 subsequently, and finally calculate final loss with Discriminator 0 and train the whole model end to end. We perform the same process for `DiscriminatorOracle()`.

## B    IMPLEMENTATION DETAILS

Table 3: Training Hyperparameters

|  | GAN | DCGAN | SNGAN | SGAN |
|---|---|---|---|---|
| Generator Learning Rate | 0.0002 | 0.0002 | 0.0002 | 0.0001 |
| Discriminator Learning Rate | 0.0002 | 0.0002 | 0.0002 | 0.0001 |
| batch size | 64 | 64 | 64 | 100 |
| Adam: beta 1 | 0.5 | 0.5 | 0.5 | 0.5 |
| Adam: beta 2 | 0.999 | 0.999 | 0.999 | 0.999 |

We implement our proposed method with Python 3.7, Pytorch=1.4.0 and Torchvision=0.5.0. We set the hyperparameters as the original implementations. We present the hyperparameters set in Table 3. We use Nashpy to compute the equilibria of the meta-matrix game.

## C    FULL TRAINING PROCESS OF 2D GAUSSIAN DATASET

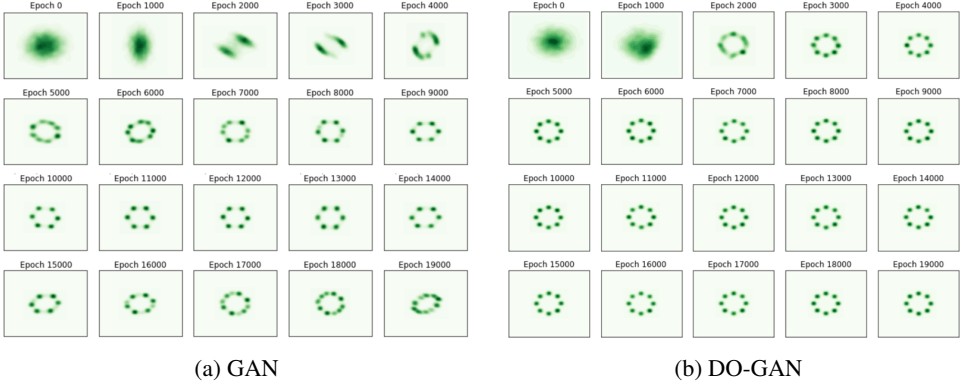

(a) GAN                                           (b) DO-GAN

Figure 6: Full comparison of GAN and DO-GAN on 2D Synthetic Gaussian Dataset

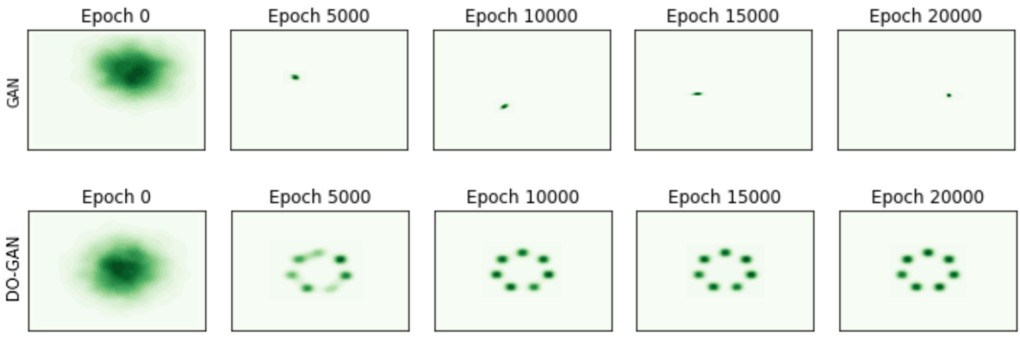

Figure 7: GAN and DO-GAN comparison with Gaussian Mixture 7 modes

Figure 6 shows the full training process of DO-GAN and GAN on 2D Synthetic Gaussian Dataset. From the results, we find that GAN struggles to generate the samples into 8 modes while DO-GAN can generate all the 8 modes of the distribution. Furthermore, DO-GAN takes shorter time (less than 5000 iterations) to identify all 8 modes of the data distribution. Moreover, we present the experiment results on 7 mode and 9-mode Gaussian Mixtures in Figure 7 and  8.

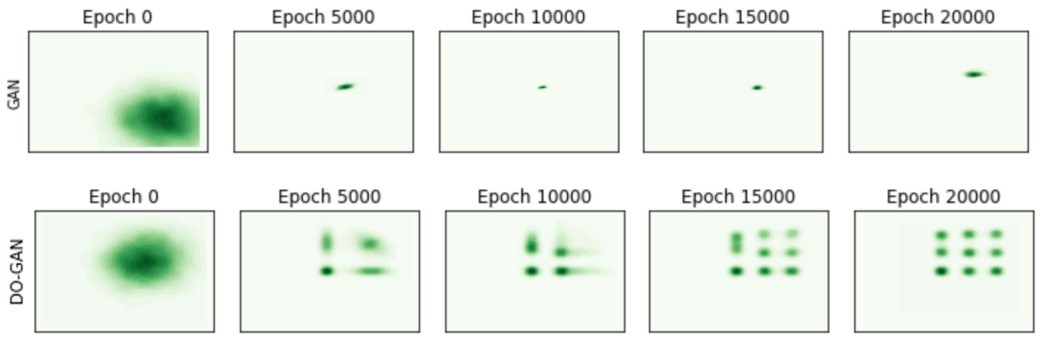

Figure 8: GAN and DO-GAN comparison with Gaussian Mixture 9 modes

## D INVESTIGATION OF SUPPORT SET SIZE

We vary the support set size $s$ to $5, 10, 15$ and record the training evolution and the running time as presented in Table 4 and Figure 9. We find that if the support size is too small, e.g., $s = 5$, the best responses which are not optimal yet have better utilities than the models in the support set are added and pruned from the meta-matrix repeatedly making the training not able to converge. However, $s = 15$ takes a significantly longer time as the time for the augmenting of meta-matrix becomes exponentially long with the support set size. Hence, we chose $s = 10$ as our experiment support set size since we observed that there is no significant trade-off and shorter runtime.

Table 4: Runtime of DO-GAN on 2D Gaussian Dataset with $s = 5, 10, 15$

| Support Set Size | Runtime (GPU hours) |
|---|---|
| $s = 5$ | $> 1$ |
| $s = 10$ | 0.5627 |
| $s = 15$ | 0.9989 |

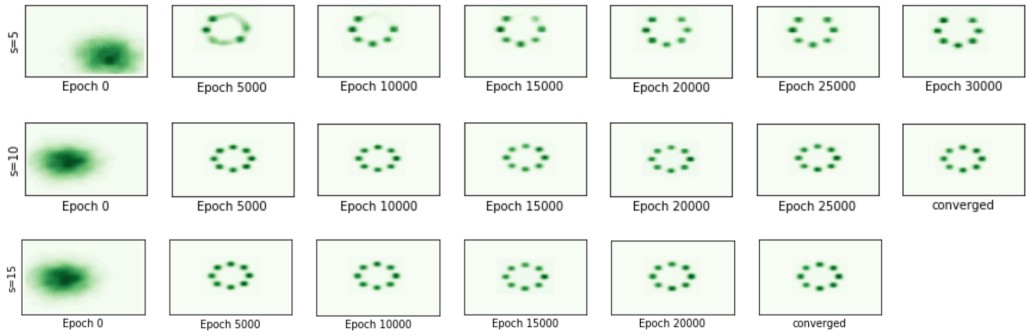

Figure 9: Training evolution on 2D Gaussian Dataset with $s = 5, 10, 15$

# E  GENERATED IMAGES OF CELEBA AND CIFAR-10

In this section, we present the training images of CelebA and CIFAR-10 datasets.

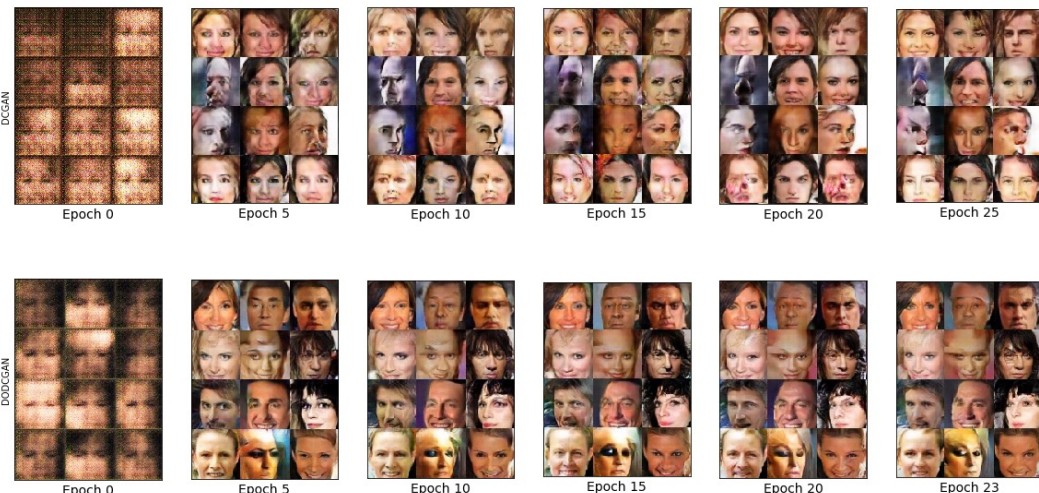

Figure 10: Training images with fixed noise for DCGAN and DO-DCGAN until termination.

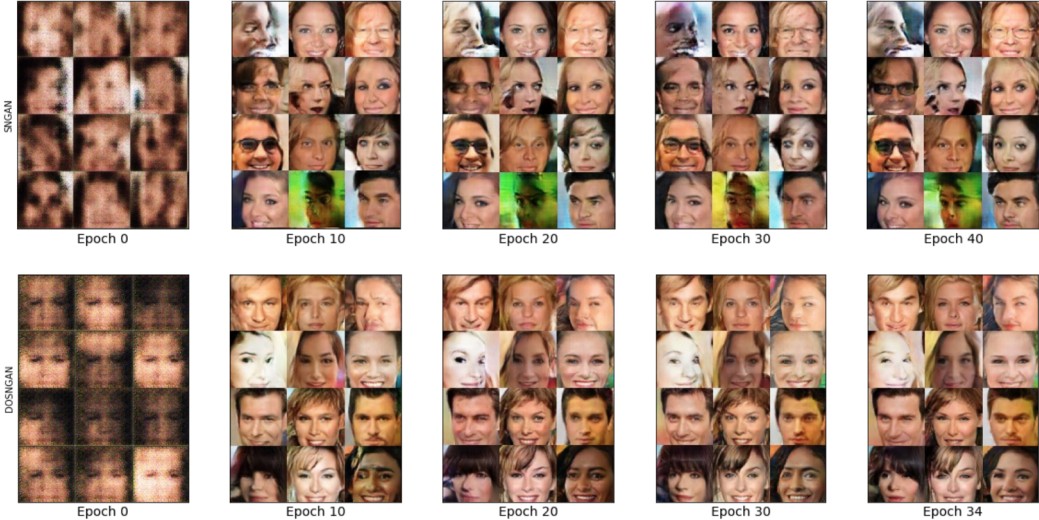

Figure 11: Training images with fixed noise for SNGAN and DO-SNGAN until termination.

Figure 10 shows the training samples of DO-DCGAN and DO-DCGAN through the training process. Figure 11 also shows those of SNGAN which is trained for 40 epochs with termination $\epsilon$ of $5 \times 10^{-5}$ for DO-SNGAN. The results show that DCGAN suffers from mode-collapse, generating similar face while DO-DCGAN can generate more plausible and varying faces. We also present the generated images of DCGAN, DO-DCGAN, SNGAN, DO-SNGAN, SGAN and DO-SGAN of CIFAR-10 dataset showing that DO-DCGAN, DO-SNGAN and DO-SGAN can generate better and more identifiable images than DCGAN, SNGAN and SGAN respectively. We present more of the generated samples from SNGAN, DO-SNGAN, SGAN and DO-SGAN on CelebA dataset in Figure 13.

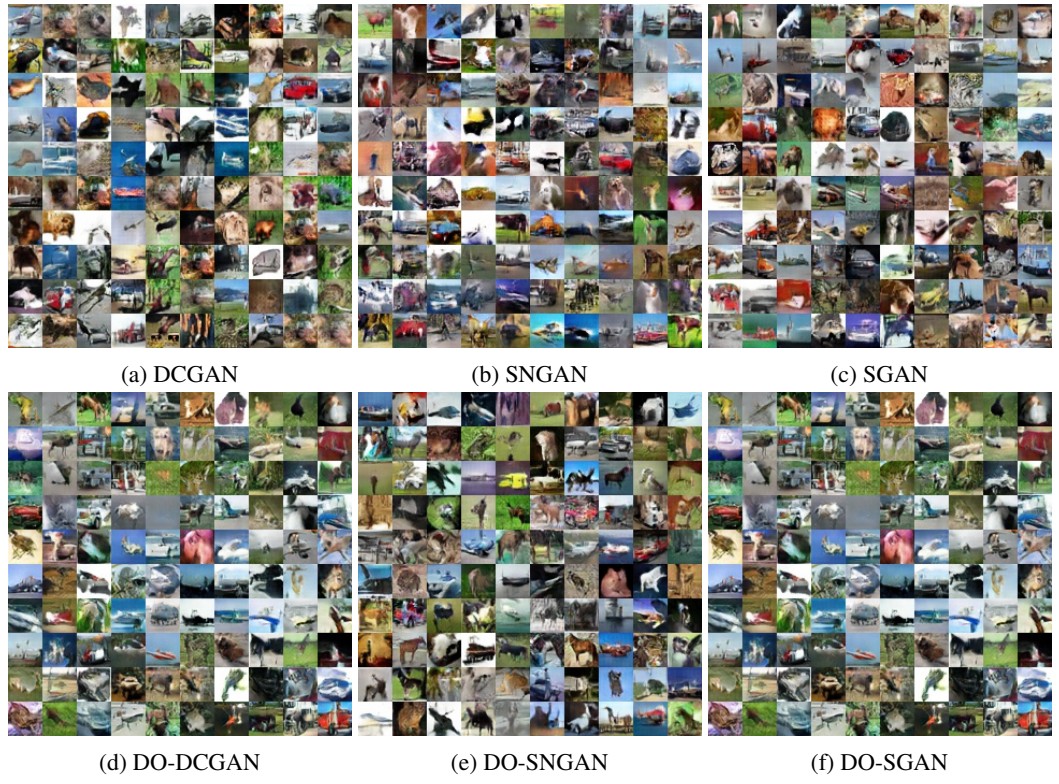

(a) DCGAN       (b) SNGAN       (c) SGAN

(d) DO-DCGAN       (e) DO-SNGAN       (f) DO-SGAN

Figure 12: Generated images of CIFAR-10 dataset

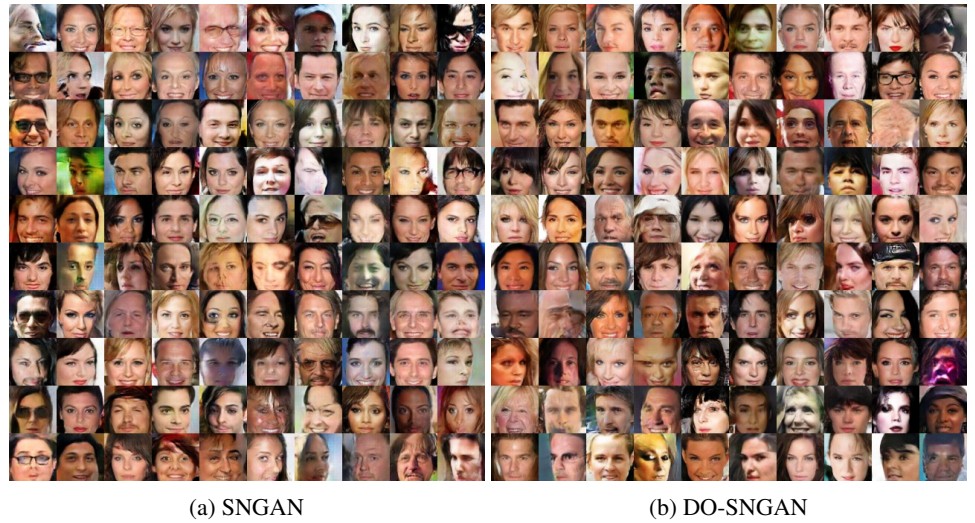

(a) SNGAN       (b) DO-SNGAN

Figure 13: Generated images of CelebA dataset for DO-SNGAN and SNGAN

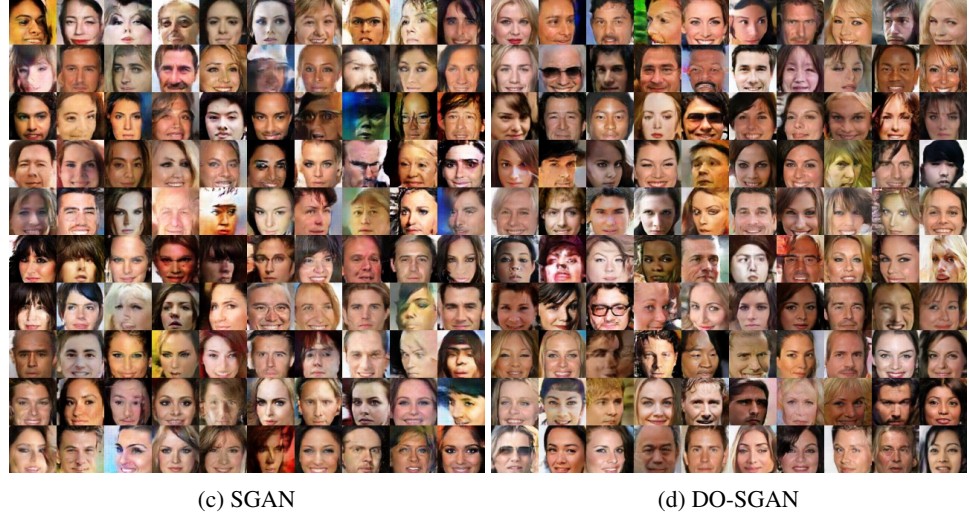

(c) SGAN                    (d) DO-SGAN

Figure 13: Generated images of CelebA dataset for DO-SGAN and SGAN

## F   FID SCORE AGAINST ITERATIONS

To compute FID score, we use Inception_v3 model with max pool of 192 dimensions and the last layer as coding layer as mentioned in (Heusel et al., 2017). We resized MNIST, CIFAR-10 generated and test images to $32 \times 32$ and CelebA images to $64 \times 64$. The FID score against training epochs for CIFAR-10 dataset is as follows:

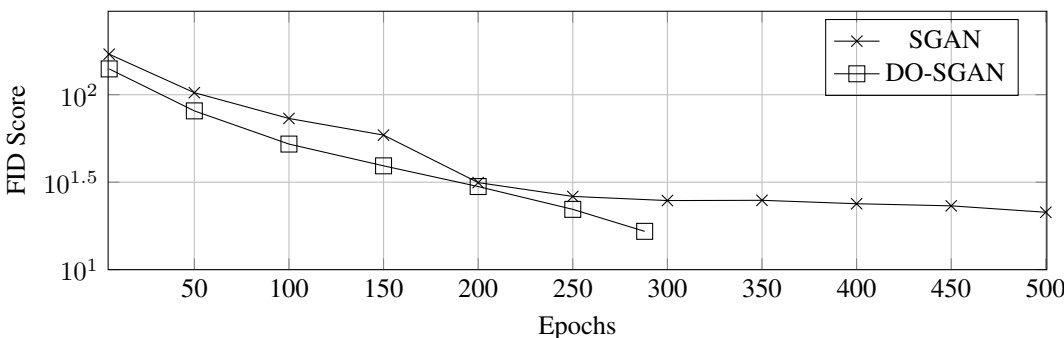

Figure 14: FID score vs. Epochs for SGAN and DO-SGAN trained on CIFAR-10

Figure 14 presents the FID score against each epoch of training for SGAN and DO-SGAN on CIFAR-10. While both perform relatively well in generating plausible images, we can see that DO-SGAN terminates early at epoch 288 and has a better FID score of 16.56 compared to 24.83 at 300 epoch until 21.284 at 500 epoch for the training of SGAN.

# G    CHOICE OF GAN ARCHITECTURES FOR EXPERIMENTS

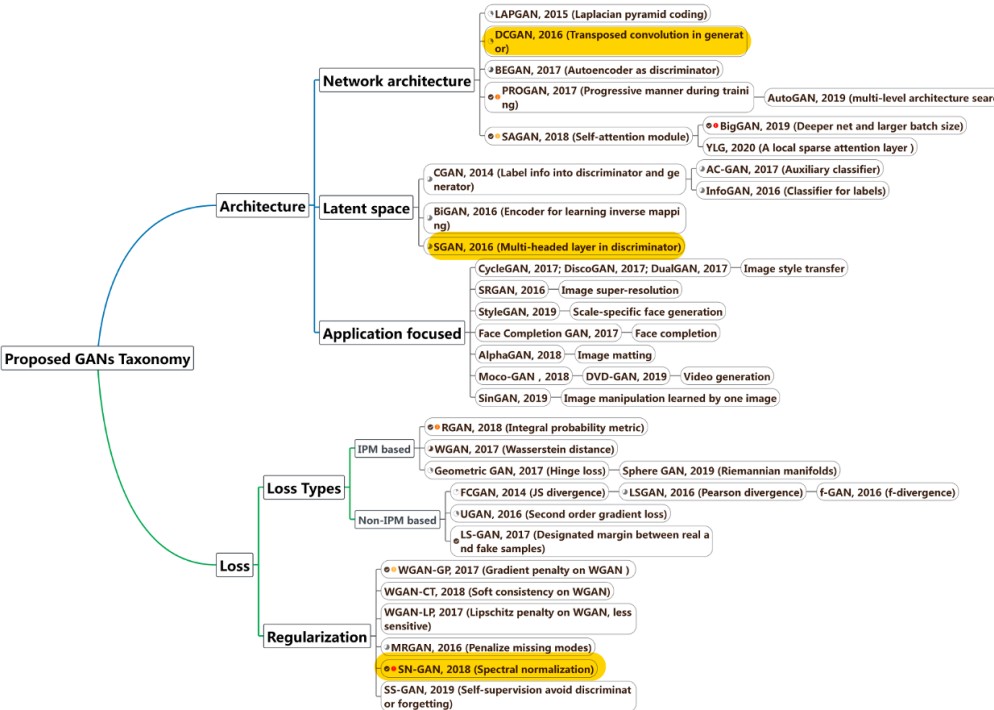

Figure 15: Taxonomy of GAN Architectures from (Wang et al., 2019)

We carried out experiments with the variants of GANs to evaluate the performance of our DO-GAN framework. We refer to the taxonomy of GANs (Wang et al., 2019) and choose each architecture from the groups of GANs focused on Network Architecture, Latent Space and Loss: DCGAN, SNGAN and SGAN as shown in Figure 15. We have also included comparisons with mixture architectures such as MIXGAN and MGAN.

## H    EXAMPLE OF META-MATRIX WITH 5 GENERATORS AND 5 DISCRIMINATORS

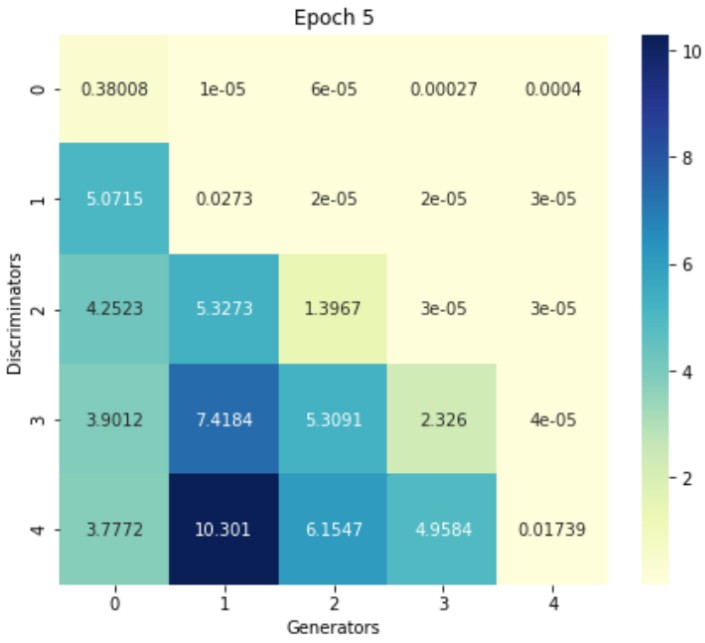

(a) Meta-matrix at epoch $t = 5$
**Meta-Strategies:** $\sigma_g^{5*} = [0, 0, 0, 0, 1]$, $\sigma_d^{5*} = [0, 0, 0, 0, 1]$
**Expected Payoff:** $U^5(\sigma_g^{5*}, \sigma_d^{5*}) = 0.017$

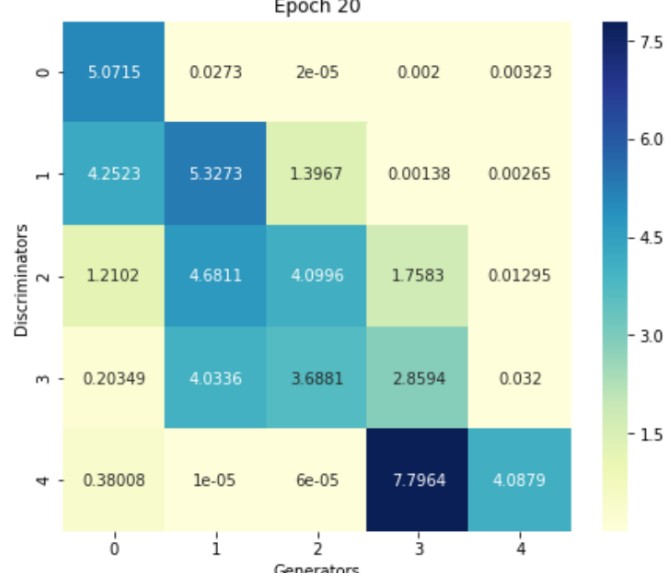

(b) Meta-matrix at epoch $t = 20$
**Meta-Strategies:**
$\sigma_g^{20*} = [0.00, 0.27, 0.32, 0.00, 0.41]$
$\sigma_d^{20*} = [0.29, 0.00, 0.32, 0.00, 0.38]$
**Expected Payoff:** $U^{20}(\sigma_g^{20*}, \sigma_d^{20*}) = 1.69$

