# OpenReview forum: "DO-GAN: A Double Oracle Framework for Generative Adversarial Networks"
_ICLR.cc/2021/Conference — Reject_

### Official Review · AnonReviewer1 · 2020-10-27
**This paper applies Double-Oracle (DO) / PSRO to training a GAN, a 2-player zero-sum game.**

**Rating:** 6
**Confidence:** 4

**Review:**

This paper applies Double-Oracle (DO) / PSRO to training a GAN, a 2-player zero-sum game. DO cannot be applied directly "out-of-the-box". Instead of an exact oracle, the generator and discriminator are trained using local gradient optimizers for a finite number of steps. Also, in DO, the meta-game matrix can grow very large and maintaining and training against a large population of neural networks is expensive, so the population is pruned throughout training.

The authors list 4 "key contributions": 1) treating D and G as players and substituting gradient optimization (Adam) for oracles to create a meta-population, 2) computing the NE of the resulting meta-game as an LP, 3) pruning the meta-population to keep the computation tractable, and 4) evaluating this approach on several GAN architectures and datasets of interest. I really only view the last one as a key contribution. The rest are contributions of prior work (McMahan, Lanctot, Cheng & Wellman).

Overall, the value of the paper lies in applying DO to GANs. I have been eager to see this tried and I view it as a necessary piece of research. The authors confirm that this approach can indeed improve performance: better Inception & FID scores on MNIST, CIFAR-10, and Celeb-A. The training speedup supported by Figures 2 and 4 is relatively meagre. The improved robustness to mode dropping in Figure 3 is good. These aren't criticisms against the paper. However, in order to draw more general conclusions on these phenomena, I would have liked to see more architectures tried. For example, not all DC-GANs are created equal. One challenge in getting GANs to work is finding the right number of layers, neurons per layer, filter sizes, strides, etc. I'm curious to learn if DO makes GANs more robust to these choices. In my opinion that would be an important contribution.

I would also like to see a discussion of training time for DO versus vanilla training. It's clear that DO is much more expensive than standard training, but by how much? Given the extra computation required, is training in 17 epochs vs 20 worth it?

As I said, there is not a lot of originality on the algorithmic side. Had the authors proposed a novel pruning scheme with superior performance on GANs, for example, that could have helped strengthen the paper.

Also, I found it interesting that the Nash distributions found in the appendix were sparse. This means that despite maintaining a population of players, we might be able to get away with a small number at test time.


Quality:
The quality of the paper is at a high enough level for ICLR.

Clarity:
The paper is generally clear.
- Please make it clear that "generatorOracle()" and "discriminatorOracle()" are approximate oracles obtained by an Adam optimizer, not "true" oracles.
- Figure 3: Should show Epoch 17 in top row as well to support your argument.
- In "Results" under Table 1, the authors state that DO-SGAN obtains much higher FID scores. I think you mean lower.

Originality:
In my opinion, the first 3 "key" contributions are really just 1: the idea of applying DO to GANs. This idea is novel, although the most interesting problems that must be solved in order to apply DO have already been tackled in previous work (PSRO - Lanctot 2017, Pruning - Cheng & Wellman 2007). The authors do not propose any significant modifications to the previous approaches. Also, the formulation of Nash of a 2-player zero-sum game as an LP is standard even within the recent DO / meta-game literature (e.g., "Open-ended Learning in Symmetric Zero-sum Games", Balduzzi 2019), so not a contribution.

Significance:
DO provides a principled training regime for GANS, that although potentially expensive, does show meaningful improvement.

---

> ### Author Response · Authors · 2020-11-21
> **Response to AnonReviewer1: Robustness + Time complexity concerns + Originality**
>
> Thank you for your valuable suggestions and comments.
>
> 1. Experiments on robustness: Your suggestion of investigating DO’s robustness of parameter choices is very good. We first focused on the fundamental architectures to show how much we can improve their performance with DO-framework. For now, we used the fixed-standard suggestions of parameter settings that worked best in the original papers i.e., same parameters with and without DO framework. We will work on it for our future direction.
>
>  2. Computational cost: Yes, DO is more expensive but meta-matrix game and pruning algorithms take polynomial time and thus, the time efficiency depends on the support set size we set which affects time efficiency of oracles, augmenting the meta-matrix with payoffs, LP and the pruning altogether. The time taken in GPU hours according to varying support set size is discussed in Appendix D: Table 2 due to the page limit, where larger support set size s=15 takes longer overall time to converge.
>
>  3. Issues with writing: The three points are fixed in the revision. Although the oracles are approximated (Adam Optimizer), the final strategy at the termination is bounded by \epsilon.
>
>  4. Originality: To apply DO to GANs, we had to tackle with a condition to convergence for GANs to be used as DO framework. As the traditional DO framework consider convergence with tractable strategy space whereas we have infinite strategy space as well as approximate best response oracle. Thus, we propose a termination check so that the best response is bounded to \epsilon. Thus, we listed it as key contribution. When we propose pruning techniques to reduce the memory load, we mentioned that the pruning method is adapted. However, we listed as contributions because to successfully apply DO to GANs, these steps are also important.

---

### Official Review · AnonReviewer3 · 2020-10-28
**Decent ideas but evaluation is lacking**

**Rating:** 4
**Confidence:** 4

**Review:**

This paper proposes to use the well-known Double Oracle methods for solving large scale games for computing the equilibrium in GANs. The main idea of a mixed strategy being a mixture over generators (and mixture over discriminator for toher player( is from Hsieh et al. The double oracle approach is shown to yield superior results on three image datasets.

Pros:
The idea of using double oracle algorithm for GAN is novel though surprising that no one has tried it earlier.

Cons:
- I feel it is very important to compare (experimentally) to Hsieh et al. ICML19. That work is also computing NE in the exact same model of mixture over generator and discriminator, which this paper borrow. Not sure why this is not done. In fact, the authors have this line about Hsieh et al "The sampling approach may be inefficient to compute mixed NE as the mixed NE may only have a few strategies with positive probabilities in the infinite strategy space" - do the authors have any evidence (theory or experiment) to support this?
- The evaluation, while exploring many GANs, is missing a few recent ones. Have the authors tried LOGAN (achieves same scores almost of CIFAR). Also, wondering why WGAN is not considered - WGAN also is a game and actually better than some of the GANs tested.
- Clearly the NE is not being computed exactly because of the many approximations (e.g., approximate best response).
- It is very surprising that only 10 pure strategies are enough in the mixed strategy support when the pure strategy space is infinite - there are results on small support for approximate NE (please cite those also). I am not sure though if it is so small, the lack of rigor here is unconvincing.
- There is no new technique in the way double oracle is used. That makes it seem like an engineering effort.
- Can some of the other approaches in PSRO framework work better than double oracle? (why do you the authors call PSRO use as "beyond games", PSRO is still being used in a game, just a very large game)
- I do not think comparing time by comparing epochs is fine - doesn't the double oracle epochs include Nash equilbrium computation. Please state these clearly.

---

> ### Author Response · Authors · 2020-11-21
> **Response to AnonReviewer3: Comparison with other approaches/architectures + Time Complexity concerns**
>
> Thank you for your valuable suggestions and comments.
>
> 1. Comparison with Mirror Descent: Hsieh et al ICML2019 proposed a Mirror Descent method which considers mixed NE solution concept to replace Stochastic Gradient Descent. GAN is considered a game with an extremely large number of pure strategies, i.e., possible parameter settings of the generator and discriminator networks. And there are only a few parameter settings which can give the convergence results. While our method also considers mixed NE solution concept, we wanted to propose a framework that can be used in various GAN architectures which originally use SGD so that their training quality is improved. Thus, we first focused on the fundamental architectures to show how much we can improve their performance with DO-framework.
>
>  2. Scope of experiments: As shown in Appendix G, we tried one from each category of GAN taxonomy and show our framework’s ability to improve the results of original architecture. For the case of LOGAN, which is the latent optimization type, we chose SGAN instead due to its architecture using stacked generators/discriminators. We chose Spectral Normalization for Generative Adversarial Networks (SN-GAN) [1] for the regularization type of GAN taxonomy. We leave WGAN since SN-GAN has been shown to perform better.
>
>  3. Approximation of NE: We added in the new draft to note that generatorOracle() and discriminatorOracle() are approximate oracle via Adam Optimizer. Although it is approximated, the final strategy at the termination is bounded by \epsilon.
>
>  4. Mixed strategy support set: We found that equilibrium at early-stage training will not be better than newly trained generator/ discriminator from the oracles as there will be better sets of pure strategies. Moreover, according to Appendix D, we can see that larger support set size of pure strategies can give better results at a cost of long training time. However, we chose 10 because it manages to converge according to our termination check and have better results than traditionally trained GAN. According to our investigation, smaller support sets are not able to converge due to infinite pure strategies.
>
>  5. Contributions: We applied the DO framework to GAN to investigate how much it can improve the performance of original GAN architectures. We also proposed the termination condition for the DO for GANs and the pruning technique to preserve the storage.
>
>  6. Usage of word: PSRO is a double oracle framework used in MARL settings. By “beyond games”, we mean that "it is also used for multiagent settings". In the revised version, we have modified as “DO is also extended to the multi-agent reinforcement learning in PSRO...”.
>
> 7. Time complexity: Computing NE of meta-matrix game and pruning algorithms only take polynomial time and thus, the time efficiency depends on the support set size we set which affects time efficiency of oracles, augmenting the meta-matrix with payoffs, LP and the pruning altogether. Apart from epochs, we also compared with GPU hours. The time taken in GPU hours according to varying support set size is discussed in Appendix D: Table 2 due to the page limit, where larger support set size s=15 takes longer overall time to converge.
>
> [1] Takeru Miyato, Toshiki Kataoka, Masanori Koyama, and Yuichi Yoshida. Spectral normalization for generative adversarial networks. ICLR, 2018a.

---

### Official Review · AnonReviewer2 · 2020-10-28
**It will be better if there are more experimental investigation.**

**Rating:** 6
**Confidence:** 3

**Review:**

This paper proposes a  new training framework for GAN, inspired by the double oracle (DO) algorithm in game theory. The authors design many mechanisms to make it possible to employ DO in GAN training. The motivation is clear, and the experimental results support the claim.

For the proposed training framework, I have the following concerns:

1. It is unclear what tasks are suitable for the proposed DO-framework. In the paper, several experiments are conducted in the laboratory and real-world environment. However, some of the more common GAN scenarios at this stage (2020) have not been proven to be suitable for DO-framework, such as high-resolution image synthesis, e.g., CelebA-HQ and LSUN scene generation [1] and conditional generation [2]. I am curious about the performance of the DO-framework in these tasks.
2. Will computational complexity be an obstacle to the application of this method? Will the proposed method require more storage space?
3. In figure.13, the DO-GAN stopped before 300 epochs. However, its FID still seems to be declining. If DO does not terminate the program, will it be better?

References:
[1] Karras, T., Laine, S., & Aila, T. (2019). A style-based generator architecture for generative adversarial networks. In Proceedings of the IEEE conference on computer vision and pattern recognition (pp. 4401-4410).

[2] Park, T., Liu, M. Y., Wang, T. C., & Zhu, J. Y. (2019). Semantic image synthesis with spatially-adaptive normalization. In Proceedings of the IEEE Conference on Computer Vision and Pattern Recognition (pp. 2337-2346).

Updates:

I have read the author's response and the comments of other reviewers. Although as an "approach as the proof of concept", limited experiments on large-scale GAN models are still necessary. Just provide evidence to prove it is feasible. This is a little disappointment for me. For my other questions, I agree with the author's response. Since I have given a positive initial rating (6), I will keep this rating. Thanks to the author's reply and AC's efforts.

---

> ### Author Response · Authors · 2020-11-21
> **Response to AnonReviewer2: Experiments + Computation/storage complexities + Termination criteria**
>
> Thank you for your valuable suggestions and comments.
>
> 1. Experiments:
> a)  We followed the experiment design in [1][2]; we believe that the current evaluation is sufficient to demonstrate the performance of our approach as the proof of concept.
> b) Our approach does not change the architecture of the original GANs. We first focused on the fundamental architectures to show how much we can improve their performance with DO-framework as a general wrapper.
> c) However, StyleGAN [3] takes 51 GPU years and is intended for large industrial applications. Given better computational facility, DO can be adapted to more complex SOTA architectures. As far as we concern, the architectures alternately update the generator and discriminator, i.e., loss and backpropagation are separate for the generator and discriminator. In such case, we can easily adapt to DO framework with oracles to provide the best responses.
>
>  2. Time Complexity: Meta-matrix game and pruning algorithms take polynomial time and thus, the time efficiency depends on the support set size we set which affects time efficiency of oracles, augmenting the meta-matrix with payoffs, LP and the pruning altogether. The time taken in GPU hours according to varying support set size is discussed in Appendix D, where larger support set size s=15 takes longer overall time to converge.
> Storage space (memory): Instead of storing in memory, we store the parameter settings of generator/ discriminator at each iteration as checkpoint files and load back while training in the oracles. Hence, the memory consumption is similar to standard GANs. The storage space for hard drive will not be much of issue as the files are relatively small.
>
>  3. Termination of DO-GAN: Yes, I agree. It might be better. However, the termination condition checks that the newly added strategy won’t improve the reward more than \epsilon and thus, we can say that although it might improve FID score, adding more strategy pairs (generator/discriminator) won’t change much to the FID score.
>
> [1] Hoang et al. "MGAN: Training generative adversarial nets with multiple generators", ICLR, 2018.
>
> [2] Huang, Xun, et al. "Stacked generative adversarial networks", CVPR, 2017.
>
> [3] Karras, T., Laine, S., & Aila, T. (2019). A style-based generator architecture for generative adversarial networks. In Proceedings of the IEEE conference on computer vision and pattern recognition (pp. 4401-4410).

---

### Official Review · AnonReviewer4 · 2020-10-28
**DO-GAN**

**Rating:** 3
**Confidence:** 4

**Review:**

Summary of paper:

This paper proposes a GAN training method that involves keeping a table of historical losses for the generator and discriminator across training iterations and calculating the loss using a double-oracle framework inspired by game theory.

Strengths:

-- To my knowledge, this is a novel approach which uses new ideas in a related field to potentially help the unstable training of GANs.

-- The background preliminaries are thoroughly explained, which is important, as much of this might be new to the target audience.

-- The idea of computing losses across iterations to save on the total number of iterations that must be performed is a good one given the high computation costs of GAN training.


Weaknesses:

-- I found this paper to be poorly written and hard to understand. The game theory language used here is not standard GAN terminology and thus made it difficult for me to follow.

-- If model weights themselves are compared/combined across iterations, for example, took me several read-throughs to understand because of unnecessary terminology like calling them "policies" and "meta-game" which are not used in work on GAN training.

-- Not all GAN loss functions are zero-sum (in fact the best performing models e.g. Big-GAN do not use a zero-sum loss), but as I see that is required for this approach, or at least is the only thing  that is considered.

-- Loss values themselves are not necessarily indicative of generative performance. There are stable equilibria that do not yield good generation, and a different equilibrium at a lower value may or may not be better. As such, only so much can be done with historical loss values.

-- I don't understand how this is supposed to help mode collapse, which is claimed as a beneficial result several times in the results section. To combat mode collapse, it will have to change the way gradients are differentiated for different points. This seems prima facie unrelated to the proposed method, and if it really is causally related, that would be valuable but this needs to be investigated with deeper experimental analysis for it to be claimed.

-- The experiments are weak. The toy example is entirely vacuous, as getting a regular GAN to match a handful of modes in 2D is quite possible with any number of small tricks that have been around for years and are not onerous.

-- The natural image results are only on CIFAR and CelebA, and are low quality at that. Qualitatively, the generated images still look bad, and the use of a DCGAN as the main baseline is misleading as the vanilla DCGAN is nobody's standard to beat anymore. Models that have achieved state-of-the-art results in the last couple years like Big-GAN should be included.



Conclusion: The motivation for this particular meta-game strategy is not very clear, and the results are not good enough to be relevant to where the current state of GANs are, and as such I vote to reject.

---

> ### Author Response · Authors · 2020-11-21
> **Response to AnonReviewer4: Writing issue regarding Terminologies + General-sum scenario + Experiment scale**
>
> Thank you for your valuable suggestions and comments.
>
> 1. Terminologies: As we are bridging the two disciplines: GAN and game theory, we try to be consistent. We agree that some reviewers are not familiar with terms from both. So, we provide a corresponding terminology table in the updated draft. We only used the term “policy” to explain the game theory concepts since it is the standard game-theory terminology but not in DO-GAN (Section 4).
>
> 2. Extension to general-sum cases: The definition of GAN is naturally a zero-sum (minmax) game where generator and discriminator competing. Even if it is general sum, our approach can be extended to a general sum case which is used in more complex architectures like Big-GAN. Computing the NE of the general sum game, and with termination condition, DO can converge to a scalable alternative of NE (alpha-rank).
>
> 3. Use of loss values: Regarding with loss, it is true that loss does not entirely determine the generative performance. We need to check with a more explanative metrics such as FID score. It might be better to use FID score in each iteration, but we will have to trade with the computation complexity. Moreover, we did not change the original architectures that we adapted which define the convergence as when the model’s loss settles down. After convergence, we evaluated the performance again with the metrics such as Inception Score and FID Score for the proper indicative of generative performance.
>
> 4. Mode collapse: DO framework helps with mode collapse since it uses mixed strategy. Mode collapse happens when generator keep using the same strategies that is best response to the discriminator. We agree that the original GAN architectures will have mode collapse since we did not change anything to the architectures.
> For example, given the first generator converge at one saddle point and suffer mode collapse (to generate “Class 1” image), the second generator will find another saddle point to get the better utility. Even if it suffers another mode collapse (to generate “Class 2” image) and same for another 8 generators, 10 different generators having different saddle points will generate different images with a probability distribution.
>
> 5. 2D experiments: Small tricks are possible to achieve generating to a few modes. But the normal GAN training depends on hyperparameters and leads to instability needing for small tricks. Our purpose is to consider a mixed NE so as not to depend on the hyperparmeter settings.
>
> 6. Experiments with Real-world datasets:
> a) While qualitative evaluation is subjective, the quantitative experiment shows that our approach can converge faster and generate competitive quality images than original architectures. We believe our method can obtain competitive or better quantitative results on inception score and FID score compared with original counterparts.
> b) This study is the first step to apply DO to GANs, providing a general framework. Our DO framework can be further explored to be adapted to better single architectures like BigGAN to improve the performance.
> c) However, for Network architecture-based taxonomy of GANs, we chose DCGAN instead of more advanced architecture such as BigGAN due to the computation facility since it is intended for large industrial applications.
> d) Given better computational facility, DO can be adapted to more complex SOTA architectures. As far as we concern, the architectures alternately update the generator and discriminator, i.e., loss and backpropagation are separate for the generator and discriminator. In such case, we can easily adapt to DO framework with oracles to provide the best responses.

---

### Author Response · Authors · 2020-11-21
**Information on the changes to the revision**

Since we adapt DO, we adopt the terminology as DO used which are the terminologies from game-theory. As we are bridging the two disciplines: GAN and game theory, we try to be consistent. We agree that some reviewers are not familiar with terms from both. So, we provide a corresponding terminology table in the updated revision.

---

### Decision · Program_Chairs · 2021-01-07
**Final Decision**

**Decision:**

Reject

**Comment:**

This paper uses the double oracle method from game theory and applies it to GANs.

This idea is interesting and Double Oracle actually seems like a good fit to train GANs. This could lead to interesting results in the future.

Reviewers disagree on the clarity of the paper, probably because the game theory vocabulary is not something that is common among the papers published at ICLR, so extra care should be taken to explain these notions.

They also point out that the method only applies to some GANs and not all (in particular the loss needs to be zero-sum).
The experimental section is too weak and the metrics used need to be changed. Results are too far from the state of the art to be convincing. The authors based their experimental setup on DCGAN: this is too old, too many improvements have made since then. Other criticisms of the experimental section include: comparison to newer methods must be made, analysis and discussion of the results must be pushed further.

Generally, the average score of the reviews is too low for acceptance. The reviewers agree that the idea is both interesting and pertinent, but this paper cannot be published as it is now. The theoretical part mostly consists in applying double oracle to GANs, and the experimental section is too weak. At least one of these parts (preferably both) must be strengthened for this paper to be impactful.